# Pilot Study of Use of Nitric Oxide in Monitoring Multiple Dental Foci in Oral Cavity—A Case Report

**DOI:** 10.3390/healthcare10020195

**Published:** 2022-01-20

**Authors:** Magdalena Wyszyńska, Przemysław Rosak, Aleksandra Czelakowska, Ewa Białożyt-Bujak, Jacek Kasperski, Maciej Łopaciński, Nour Al Khatib, Małgorzata Skucha-Nowak

**Affiliations:** 1Department of Dental Materials, Division of Medical Sciences in Zabrze, Medical University of Silesia in Katowice, 15 Poniatowskiego Street, 40-055 Katowice, Poland; ebialozyt@sum.edu.pl; 2Specialist Dental Practice Przemysław Rosak, 13 Piłsudskiego Street, 41-300 Dąbrowa Górnicza, Poland; rosakowski@o2.pl; 3Department of Dental Prosthetics, Division of Medical Sciences in Zabrze, Medical University of Silesia in Katowice, 15 Poniatowskiego Street, 40-055 Katowice, Poland; aczelakowska@op.pl (A.C.); protstom@sum.edu.pl (J.K.); 4Department of Periodontal Diseases and Oral Mucosa Diseases, Division of Medical Sciences in Zabrze, Medical University of Silesia in Katowice, 15 Poniatowskiego Street, 40-055 Katowice, Poland; mlopacinski@sum.edu.pl; 5Student of 4th Year Dentistry Program, Student Scientific Society in Department/Institute of Prosthetic Dentistry and Dental Material Sciences, Division of Medical Sciences in Zabrze, Medical University of Silesia in Katowice, 15 Poniatowskiego Street, 40-055 Katowice, Poland; noorak716@gmail.com; 6Department of Dental Propedeutics, Division of Medical Sciences in Zabrze, Medical University of Silesia in Katowice, 15 Poniatowskiego Street, 40-055 Katowice, Poland; mskucha-nowak@sum.edu.pl

**Keywords:** periimplantitis, nitric oxide, inflammation proces, inflammation markers

## Abstract

Background: The most common cause of implant loss and deteriorating restoration aesthetics is infection and chronic inflammation of the tissues around the implants. Inflammation in the oral cavity, confirmed by clinical and histopathological examination and determination of exhaled nitric oxide, is a situation which may cause the complications on the whole human body. Elimination of the patology in the oral cavity in some cases is the only resonable treatment. The aims and objectives of our work is to present a gradual treatment of advanced infalmmation and present huge reduction stamp of inflammation measured with marker nitric oxide (NO) in exhaled air. Materials and Methods: Simple treatment containing elimantion of pathology in the oral cavity was conducted. Patient that came to the dental practice suffered from the inflammation caused by lack of proper hygiene. First aid in this situation was to eliminate the inflammation which may affect negatively for general health. At first visit full hygienization was performed, at the second visit roots of abutment teeth and implants were removed under local anesthesia along with cystic changes. Results: The hygiene precedures and extraction of the unsteady inflammationprosthetic restorations significantly decreased the level of NO in exhaled air. Conclusions: During the examination of the patient coming to the dental practice great attention should be paid to the coexistence of pathologies related to the oral cavity. Omission of a dental examination and possible elimination of odontogenic foci may affect the implication of the results of general diagnostics and subsequent treatment. Measuring the level of NO on exhaled air seems to be useful diagnostic method.

## 1. Introduction

Modern implant–prosthetic restorations enable long-term treatment effects, even in cases of osteoporosis or diabetes. However, despite the high efficacy of treatment at the level of 89–90%, failures still occur [1]. The most common cause of implant loss and deteriorating restoration aesthetics is infection and chronic inflammation of the tissues around the implants. This leads to the resorption of the bone tissue and consequently to recession of gingiva. This is accompanied by a gradual exposure of the implant surface and its threads. The most common causative factor is bacterial infection, or rather an imbalance between opportunistic flora and the host’s immune mechanisms. The following factors play a significant role: systemic diseases, genetic factors, occlusion disorders, social factors: stress, smoking, poor oral hygiene [1,2]. The other factors that may cause failure in treatment are: insufficient quantity and quality of bone tissue—in this case, the error occurs at the diagnostic stage. The correct determination of the quantity and quality of the bone in the selected implantation site is one of the issues that determine the success of treatment, too early load of the implant with prosthetic work—before placing the prosthetic restoration. Improper surgical technique—an incorrect way of inserting the implant may decide about the failure of the procedure. Too much insertion force, too deep screw insertion, excessive screw rotation, bone overheating resulting from too high torque of the drill—all these failures may result in implant loss or partial treatment failure. The success of the implant placement procedure is a derivative of many factors: the skills and knowledge of the operator, the patient’s health condition and the individual predispositions of his body. It is also extremely important to maintain good oral hygiene both before and after the procedure. If the doctor’s instructions are not followed, infection and inflammation of the tissues around the implant may occur, which causes the loss of osseointegration. Inflammation is usually caused by an attack by bacteria in the plaque that is not removed effectively enough. Research shows that insufficient oral hygiene is the main reason for implant loss [3,4]. Use of dental implants is a proven method of dental treatment in case of missing teeth in various clinical situations. According to the literature the implantation procedure success rate after 16 years was 82.9% [5]. Succes in treatment depends on precise indications and contraindications both general and local. The condition for successful treatment is continuous monitoring of the patient through control visits, during which the doctor checks the hygiene of the oral cavity and tissues around the implants. Inflammation and damage to the soft and hard tissues surrounding dental implants can be defined as mucositis or periimplantitis [6,7,8]. Thus, the boundary between these states is often fluid and incompletely expressed clinically [9]. Mucositis is a reversible inflammatory process of the soft tissues surrounding the implant of bacterial origin, with redness, swelling and bleeding when examined with a probe [6,7,8,9,10]. These are typical symptoms, but not always clearly visible. Moreover, bleeding on probing (BOP) may be a marker of periimplant disease, but there is still insufficient evidence of its prognostic value [11]. Unlike mucositis, periimplantitis is a progressive and irreversible process involving soft and hard tissues surrounding the implant, accompanied by bone resorption, reduced osseointegration, deepening of the gingival pockets and suppuration. Bleeding on examination, bone loss and deep gingival pockets may have reasons other than inflammation, eg too deep implant placement [12]. Moreover, the condition of soft and hard tissues surrounding the implant may be influenced by: the type and shape of the implant, the type and material of the abutment, the type and material of the prosthetic superstructure [11]. Apart from the clinical symptoms, the signs of inflammation are also visible in the histopathological picture. The histopathological picture of the periimplantitis focus has a microscopic picture of the inflammatory infiltrate similar to that seen in periodontitis. The tissue is inflamed by macrophages, lymphocytes and plasma cells. Bone necrosis may appear. The inflammatory reaction is not always clear [13,14,15]. Inflammation in the oral cavity, manifested by the clinical and histopathological picture, is also associated with an increase in inflammatory markers. One of the state markers of ongoing inflammation is nitric oxide (NO) [16,17]. In the diagnosis of diseases of various origins, blood tests, urine tests, and histopathological tests are most often used, while determining the concentration of NO in the exhaled air is less popular. Measurement of the concentration of NO in the exhaled air is used, among others, in the examination and monitoring of the treatment of patients with inflammation in the oral cavity, respiratory tract and digestive system. The device used to measure NO in the exhaled air is the Niox Mino^®^ (Aerocrine AB, Solna, Sweden) [18].

NO is formed by the enzyme nitric oxide synthase (NOS) due to oxidation L-arginine to L-citrulline and nitric oxide. The NO molecule has the structure of a free radical. NO is a colorless and inorganic gas with versatile action, it participates in both pathological and physiological processes. Active nitrogen compounds have a cytotoxic as well as immunoregulatory effect. The NO molecule can react with oxygen to form the NO^2−^ radical, which in turn produces nitrites (NO^2−^) and nitrates (NO^3−^). In the course of further chemical transformations, peroxynitrite is formed, which after the so-called protonation creates peroxynitrous acid. Eventually, the hydroxyl radical and nitrogen dioxide are formed. In inflammation, NO is produced continuously for hours or even days. Its formation is closely related to the participation of induced nitric oxide synthase (iNOS). As a result of this process, the enzyme cyclooxygenase (COX) is activated, which stimulates the production of reactive oxygen species and pro-inflammatory prostaglandins. As a consequence of these processes, the blood vessels dilate excessively, which become permeable to albumin, and this causes edema. During septic shock, where there is an overproduction of nitric oxide, the heart begins to fail, blood pressure drops sharply, and the response to vasoactive agents decreases. Physiologically, NO is formed by endothelial nitric oxide synthase (eNOS) in endothelial cells and affects the smooth muscles of blood vessels. It is a regulator of blood flow and pressure [19,20,21]. In the inflammatory process, blood cells (neutrophils, monocytes) and endothelial cells of blood vessels activated by antigens (bacteria, viruses, parasites, carcinogens) and endothelial cells release many pro-inflammatory mediators, the overexpression of which causes tissue damage. These mediators are numerous cytokines, such as the following interleukins: IL-1beta, IL-2, IL-6 as well as TNF-alpha and IF-gamma, which stimulate white blood cells, mainly macrophages, to produce significant amounts of NO through long-term activation of iNOS (synthase nitric oxide). At the same time, the induction of NADPH and xanthine oxidase takes place, and the resulting superoxide anion releases selectin activators (histamine, thrombin) from mast cells, facilitating the first contact of leukocytes with endothelial cells and the process of their rolling along the vessels. The simultaneously released adhesion activators, platelet activating factor (PAF), leukotriene B4 and C5A enable the adhesion of leukocytes to endothelial cells and their subsequent migration to the inflammatory site. Superoxide anion (O^2−^) can also form a very toxic peroxynitrite in reaction with NO [22]. The significant toxicity of peroxynitrite (ONOO^−^) can be reduced by the enzyme superoxide dismutase (SOD), which breaks down the superoxide anion, i.e., reduces its concentration. Nitric oxide also has a positive effect. It has the ability to inhibit the production of the above-mentioned superoxide anion by influencing the O^2−^ generating enzyme, i.e., NADPH oxidase, through neutrophils. NO reduces the adhesion of neutrophils to the endothelial cells of the coronary vessels, and also reduces the breakdown of mast cells located near blood vessels, which are the source of inflammatory mediators. This reduces the spread of the inflammation mechanism and the leukocyte migration towards the inflammatory focus. All these events lead to the conclusion that NO controls the first stage of the inflammatory process, and thus prevents damage to the body’s tissues. Exhaled NO is a sensitive indicator of the inflammatory process, responding rapidly to treatment or exacerbation of the disease. According to the literature, elevated NO concentration is very well compatible with other inflammatory indicators assessed in biopsy material, fluid obtained from bronchoalveolar lavage, or induced sputum [23,24].

Therefore in the authors’ opinions, it is reasonable to determine the level of NO to attempt monitoring the course of inflammation in patients with pathological changes oral cavity.

## 2. Aims and Objectives of the Study

The aim of this work is to present a gradual treatment of advanced infalmmation and present huge reduction stamp of inflammation measured with marker nitric oxide (NO) in exhaled air. Described case report reveal how important is the oral health in general diagnostics and present additional diagnostic method.

## 3. Materials and Methods

Patient N.P. was 61 years of age, and came to the dentist’s office for dental-prosthetic treatment. The first visit consisted of a precise interview (general and dental health), intraoral examination, using basic dental instruments and WHO periodontal probe. The mobility of prosthetic restorations in three planes, was measured in manual examination, third degree according to the PTV scale (Periotest value). The radiological examination was conducted before the treatment. The NO concentration was tested with the Niox Mino^®^ device (Aerocrine AB, Solna, Sweden). Niox Mino^®^ is a portable medical device for measuring fractional exhaled nitric oxide (FeNO), providing results in parts per billion (ppb—the number of NO particles in one billion gas particles, i.e., 1 ppb is equal to 1 nl/L). The device has an NO (scruber NO) filter, which eliminates nitrogen oxide present in the atmosphere. The flow control (NIOX flow control) maintains the expiratory air flow rate at 50 mL/s regardless of the patient’s abilities. Each measurement is checked and an automatically performed test checks the isoline level. This ensures repeatability and correctness of measurements. The so-called dead space thanks to specially designed filters and the placement of the sampling port as close to the mouth as possible. It is a simple, non-invasive, patient-friendly diagnostic test useful in patients with bronchial asthma, which consists in inhaling air from the device and then blowing air into the device evenly. The results are displayed on the monitor of the device in 2 min. The patient receives a disposable mouthpiece to conduct the examination [18]. At the end of this visit hygienization procedures were conducted. At the second visit measurement of NO in exhaled air was conducted. Next all unsteady prosthetic restorations were exctracted under local anesthesia. The removed cyst fragments were sent for histopathological examination. Patient was treated with temporary removable dentures. During last visit the control and last measurement of NO in exhaled air were conducted. Unfortunately the patient never came back after that visit.

## 4. Case Report

Patient N.P. was 61 years of age, came to the dentist’s office for dental-prosthetic treatment. 

### 4.1. First Measurement of NO in Exhaled Air

At the first visit, in the interview, the patient did not mention any chronic general diseases, did not take medications on a permanent basis, and did not use any stimulants (nicotine). The prosthetic restorations that he was using were made eight years earlier in another office and since then the patient has not used the help of a dentist. Clinical intraoral examination revealed abundant plaque, chronic generalized gingivitis and periodontitis, swelling and redness of the marginal gingiva tissues near the natural teeth and implants, blood exudate during probing of the gingival fissure using a WHO periodontal probe in the area of impantomosts 13–18 and 35–37 were found. Advanced periimplantitis, significant mobility of prosthetic restorations in three planes, clearly perceptible in manual examination, third degree according to the PTV scale (Periotest value). During diagnostic examination, the dental practitioner revealed irregularities in the design of the prosthetic structure, consisting in the connection of the patient’s own teeth and implants as prosthetic pillars of the bridge, which could in his opinion also contribute to the pathological mobility of the prosthetic restoration (Figure 1).

The radiological examination revealed the presence of an implant bridge in section 13–18 (on implants and natural teeth), where significant bone loss is visible, teeth in section 12–27 with numerous cavities and fillings, in the area of 35–37 the implant bridge, around which there was significant bone loss and 35 and 37 implant fractures on the border of bone atrophy. The remaining dentition in the mandible has numerous fillings and caries (Figure 2). The NO concentration was tested with the Niox Mino^®^ device. The value of NO concentration in the exhaled air was 126 ppb.

At the first visit, and after the examination, full hygienization was performed, i.e., ultrasonic removal of dental plaque, sandblasting and polishing of the teeth. Removal of unsteady fixed prosthetic restorations was postponed until the next visit. The instruction of keeping proper hygiene was conducted. The patient was advised to improve hygiene of the oral cavity.

### 4.2. Second Measurement of NO in Exhaled Air

The second visit took place three weeks later, on the same day and similar hours. The patient’s history showed features of general health, did not take any medications, and there were no other acute or chronic diseases. An improvement in the condition of the oral mucosa was noticed in the intraoral examination. Gums around the patient’s own teeth without signs of swelling, pink, shiny, not bleeding during probing. Persistent inflammation in the area of implant bridges 13–18 and 35–37 of less intensity than during the first visit. The value of exhaled nitric oxide concentration determined during the second visit was 72 ppb, which is a 42.86% decrease of the level of inflammation marker (Table 1). After the examination and control of the oral cavity condition, the extraction of unsteady, permanent prosthetic restorations on implants was started. Fixed upper and lower prosthetic restorations, roots of abutment teeth and implants were removed under local anesthesia along with cystic changes (Figure 3a,b). The removed cyst fragments were sent for histopathological examination.

### 4.3. Histopathological Examination

The results of the histopathological examination showed a connective tissue lesion with the features of a cyst with a cavernous space lined with a multilayered squamous epithelium, focally changed with necrosis. In the wall of this cyst there is a confluent, massive chronic inflammatory infiltrate consisting of the predominant number of cells from lymphocytes and plasma cells, also visible hyperemia and granulation tissue in the tissues. The preparations also present loose lying connective tissue fragments with focal presence of epithelial cover, with single bone trabeculae, chronic type inflammatory infiltrate with high intensity, with granulation tissue and small foci of necrosis (Figure 4 and Figure 5).

After seven days, the patient returned to have his sutures removed. Conservative treatment of tooth decay was carried out at subsequent visits.

### 4.4. Third Measurement of NO in Exhaled Air

At the follow-up visit after a month from second measurement, third measurement of exhaled NO was conducted, in the same day and similar hours. The patient’s history showed features of general health, did not take any medications, and there were no other acute or chronic diseases. The level of NO was measured and it was 31 ppb which is 75.4% decrease according to the first visit. Patient was treated with temporary removable dentures. Unfortunately, the patient has not reported to continue the prosthetic treatment since last visit, which confirmed the lack of cooperation between the doctor and the patient and in consequence the failure of the treatment.

## 5. Results

Treatment conducted in this case relied on the removing the inflammatory process. The hygiene precedures and extraction of the unsteady inflammated prosthetic restorations si gnificantly decreased the level of NO in exhaled air (Table 1).

## 6. Discussion

The most common cause of implant loss inflammation patholgy leading to bone tissue resorption [7,25]. Often the causes of implant loss are also early bacterial infections during the implantation procedure. The most common pathogen causing them is *Staphylococcus aureus* [26,27]. However, they cannot be confused with the concept of periimplantitis, which is associated with the colonization of periimplant tissues by the opportunistic flora of the oral cavity. The healthy tissue surrounding the implant is characterized by a large number of granular cells, a small amount of aerobic and anaerobic bacteria, Gram-negative species and pathogens responsible for periodontitis. In the case of periimplantitis, the conventional bacterial flora characteristic of periodontitis is dominant. However, perimplant lesions include not only bacteria associated with periodontitis, but also bacteria of the species *Staphylococcus, Enterococcus* and *Candida* [28]. It should be noted that the bacterial flora is not the only causative agent of periimplantitis. An important role is also played by the polymorphism of the IL-1β genotype, nicotinism, age of the patient. The IL-1β positive genotype is characterized by increased IL-1 synthesis, which results in a 2.7 times higher risk of loss of natural teeth, but in the case of loss of implants, the results are not so unambiguous. There is also a link between the increased level of pro-inflammatory factors, such as lactoferrin and elastase as well as IL-1β, and the occurrence of periimplantitis [29,30]. Zitzmann et al. estimated the incidence of periimplantitis development in patients with a history of periodontitis as almost six times higher than in patients without a history of periodontitis. After 10 years, the symptoms of the inflammatory process occurred around 10–50% of the implants [31,32]. Mombelli et al. found periimplantitis in 20% of all implant patients and in 10% of all implanted implants. Although this percentage should be interpreted with caution due to the diversity of the analyzed studies, the researchers emphasize the fact that the bone remodeling process often results in marginal bone loss in the first weeks after abutment insertion, which should not be considered as periimplantitis [11]. This led to the formulation of a recommendation to make a radiograph after fixing the superstructure and use it as a starting point for the subsequent assessment of bone loss around the implant. Wallowy et al. have shown that the presence of periodontal diseases or smoking, increases the risk of periimplantitis almost five times [33,34,35]. Smoking increases the rate of bone loss by 0.16 mm per year and is a major systemic risk factor [36]. Smokers show a lower extent of osseointegration and a worse condition of oral hygiene in the vicinity of implants [37]. It is widely accepted that nicotine negatively affects the results of parameters assessing the effects of oral treatment procedures, although not all previous studies have found positive correlations between periimplantitis and smoking [38,39]. At the microscopic and molecular level, fundamental differences can be identified between periimplant tissues and unchanged periodontal tissues. Due to the reduced vascularity and the parallel arrangement of collagen fibers, periimplant tissues are more sensitive to inflammation than periodontal tissues. This phenomenon can be confirmed immunohistochemically by demonstrating the increased production of factors involved in the formation of the inflammatory infiltrate, such as nitric oxide, lymphocytes, and neutrophils. Moreover, similarly to periodontitis, lesions may show an almost 10-fold elevated concentration of matrix metalloproteinases (MMP), such as MMP-8, which can be used for diagnostic purposes [40,41]. Berglundh et al. analyzed the histological picture of periimplantitis foci. A massive inflammatory infiltrate was observed, expanding apical, dominated by plasma cells. Large amounts of polynuclear cells were present not only in the pocket epithelium, but also in the perivascular zone [42]. In their research, Gualini and Berglundh compared the picture of periimplant mucositis and periimplantitis lesions. The latter were larger and contained more B cells (CD19+) and elastase producing cells [43]. Zitzmann N.U. et al., studied foci of experimentally induced mucositis in humans in the vicinity of the implant and the rest of the oral cavity (mucosa response to plaque accumulation). Increased inflammatory infiltration was observed in both regions, with no statistically significant differences, although the host’s response was more marked outside the implantation regions [44]. Prevention of periimplant tissue diseases should begin with appropriate planning, which takes into account the individualized assessment and minimization of risk factors (smoking, compliance, oral hygiene, periodontal disease, systemic diseases), ensuring the optimal condition of soft and hard tissues, choosing the best model for a given situation of the implant, followed by maximally atraumatic surgery and regular checkups with periodontal assessment. Treatment of periimplant tissue infections includes conservative (non-surgical) and surgical procedures. Depending on the severity of the periimplant disease (mucositis, moderate or severe periimplantitis), conservative treatment alone may be sufficient, or a step-by-step approach with a surgical procedure followed by conservative treatment may be necessary. Most of the published recommendations for treatment strategies in periimplantitis are mainly based on the methods used in periodontitis. The reason for this is a similar way in which the bacterial colonization of the surface of the natural tooth and the implant takes place, it is widely accepted that the bacterial biofilm plays an analogous role in the development of inflammation of the tissues surrounding the implants [45]. Unfortunately, in the above mentioned case the inflammation was too advanced. The treatment described above consisted of the elimination of inflammatory foci by sanitation of the oral cavity. The previous treatment failure in the described patient, in authors’ opinion was caused by mistakes in designing the prosthetic construction and incorrect oral hygiene. The lack of follow-up visits made it impossible to detect the developing pathology earlier and to start appropriate therapeutic and prophylactic treatment. Pathological processes in the oral cavity, such as caries, periodontitis or inflammation related to periimplantitis, are the body’s reaction to the infection. In the inflammatory process, blood cells (neutrophils, monocytes) and endothelial cells of blood vessels activated by antigens (bacteria, viruses, parasites, carcinogens) and endothelial cells release many pro-inflammatory mediators, the overexpression of which the tissue damage is caused by [46,47]. In the last 20 years, a new method for monitoring the inflammatory process within the lungs has been recognized—the assessment of the concentration of nitric oxide (NO) in the exhaled air. It has been observed that the concentration of exhaled nitric oxide is proportional to the degree of inflammation found in the bronchial wall, eosinophilia in induced sputum and bronchial hyperreactivity. The increase in NO concentration is associated with the exacerbation of inflammation, and its decrease—with the anti-inflammatory treatment used in asthma. For this reason, it is a method recommended for monitoring the course of this disease [48,49,50,51,52,53,54,55,56]. Changes in the exhaled NO concentration are not specific only to asthma. They are also found in other diseases, such as: allergic rhinitis, eosinophilic bronchitis [57,58]. The other pathologies in which NO measurement can be useful are: diagnosis of eosinophilic airway inflammation, determining the likelihood of steroid responsiveness in individuals with chronic respiratory symptoms possibly due to airway inflammation, to indicate that eosinophilic inflammation and responsiveness to corticosteroids are less likely [59,60,61]. The literature reports that the concentration of NO was tested in preparations of saliva, gingival pocket fluid, a section of the inflamed mucosa, and a tissue sample from the mass of the neoplastic lesion. Vanessa G.S. Reher et al. studied the level of NO in saliva using the Griess reaction. Based on the conducted research, the authors found that the concentration of NO in saliva increases along with increasing periodontitis [62]. D.F. Lappin et al. took sections of inflamed tissues, subjected them to immunohistochemical tests, and compared them with healthy tissue samples. High levels of NO in inflamed tissues are associated with tissue destruction and cell death. They showed a clear role of NO in initiating the chain of immune response in the case of infection of periodontal tissues [63]. Kazuyuki Shibata et al. observed an increase in NO levels in aggressive localized periodontitis. In this pathological state, the function of neutrophils and the chemotaxis process, with which NO and NO synthase are related and disturbed [64]. The study of NO concentration in gingival tissues was also carried out by Michiko Hirose et al. The research material was collected from the bottom of the gingival pocket and from the gingival surface. The authors showed an increase in the concentration of interleukin-6 and NO in periodontopathies. The inflammation of periodontal tissues, including the gums, is initiated by the colonization of pathogenic bacteria in the oral cavity. Cytokines and NO released from inflammation play a significant role in the process of the entire cascade of immune response [65,66].

The advantage of this diagnostic method is that it is simple to conduct and the results are obtained very fast. Unfortunately, this type of mesurement is suitable only for patients with good health not suffering from any general diseases and not using nicotine which could affect the measurement parameters. The conducted pilot study proved that the use of NO measurements in the diagnostics of respiratory system diseases should not be performed without assessing the condition of the oral cavity. Only the patient’s compliance with the hygiene instruction carried out at the first visit led to an improvement in the local condition of the marginal periodontal tissues observed in the oral cavity examination and a decrease in the NO level in the exhaled air measured during the second visit. Consistent sanitation of the oral cavity aimed at removing inflammatory foci led to further decrease of the examined parameter (Table 1). Omission of a dental examination and possible elimination of odontogenic foci may affect the implication of the results of general diagnostics and subsequent treatment. According to the authors, it is recommended to continue research in the above-mentioned area.

## 7. Conclusions

According to this case report, it seems that the measurement of NO concentration in the exhaled air is a beneficial diagnostic method. The treatment conducted in this case review showed huge impact of inflammation on level of NO in exhaled air, which increases adequately. That confirms that NO in exhaled air seems to be valuable and easy to conduct inflammation marker. However, when assessing and interpreting the results of level of NO in exhaled air among patients coming to medical clinics attention should be paid to the coexistence of pathologies related to the masticatory organ. There is a need for dentists and medical practitioners, e.g., pulmonologists to work together in the use of exhaled nitric oxide measurement.

## Figures and Tables

**Figure 1 healthcare-10-00195-f001:**
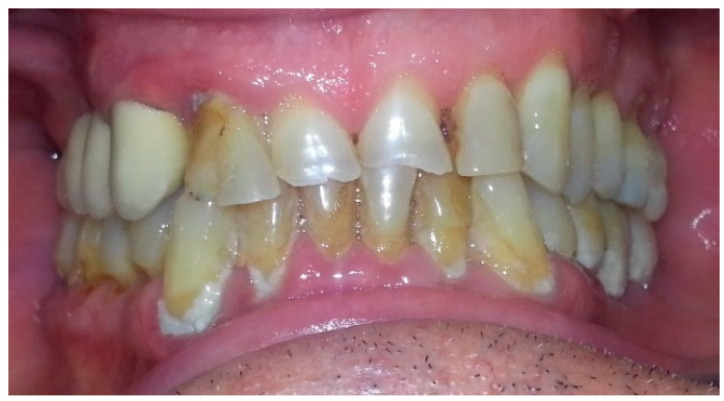
Frontal view of patient intraoral in first visit.

**Figure 2 healthcare-10-00195-f002:**
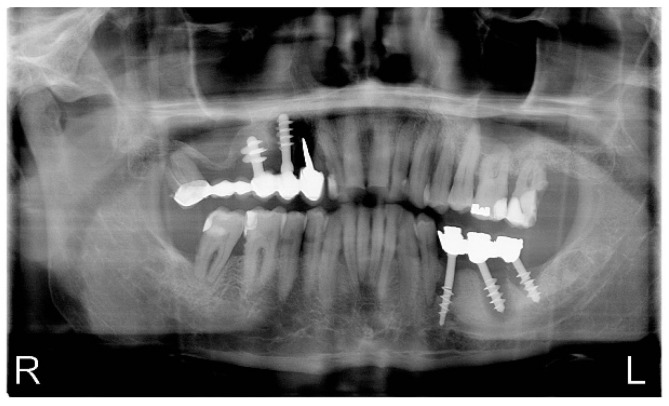
Preoperative panoramic x-ray.

**Figure 3 healthcare-10-00195-f003:**
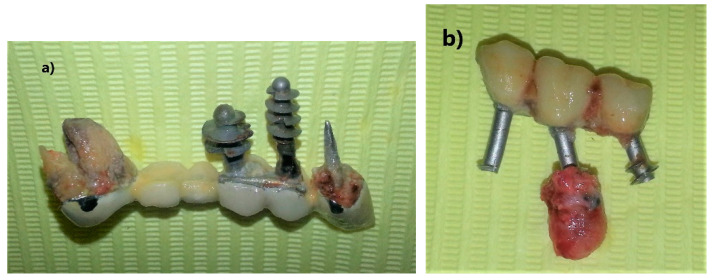
(**a**) Extracted upper fixed restoration; (**b**) extracted lower fixed restoration.

**Figure 4 healthcare-10-00195-f004:**
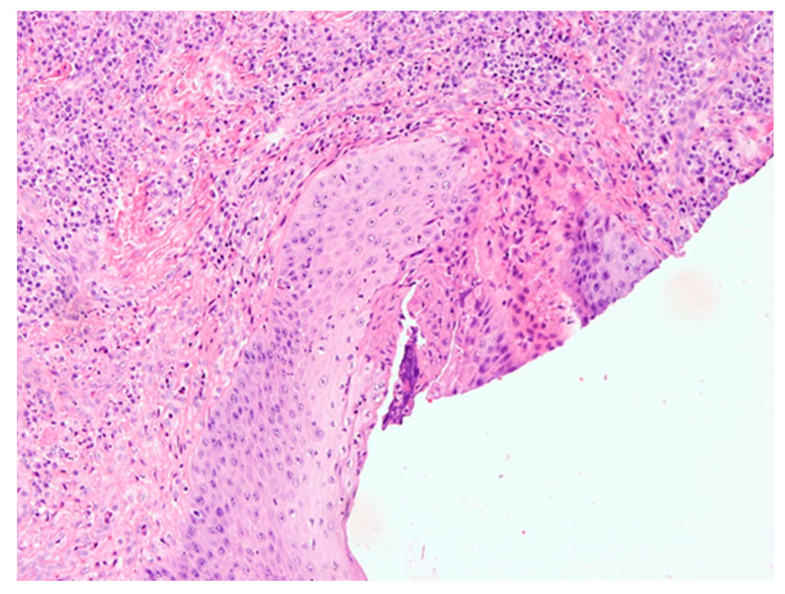
A cystic section with a squamous lining with a massive mixed-cell inflammatory infiltrate composed of lymphocytes, plasma cells, with an admixture of neutrophilic and eosinophilic granulocytes. H + E staining.

**Figure 5 healthcare-10-00195-f005:**
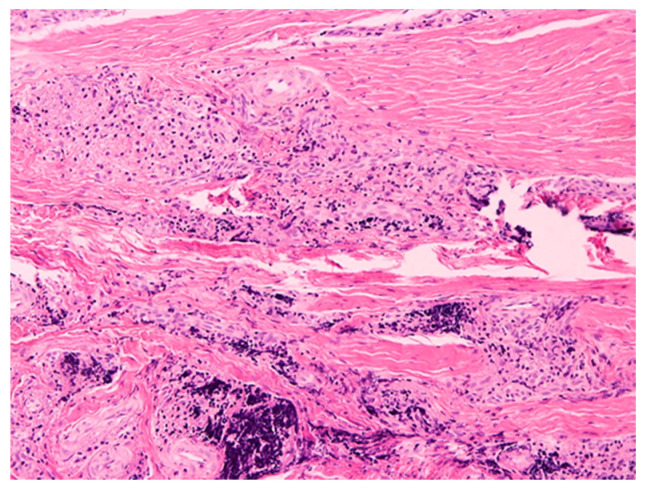
Cyst wall with massive chronic inflammatory infiltrate. H + E staining.

**Table 1 healthcare-10-00195-t001:** Change in the value of NO concentration in the exhaled air.

l.p.	* Weeks	Level of Exhaled NO (ppb)	* Drop in Level of Exhaled NO (%)
1.	3	72	42.86%
2.	7	31	75.4%

* In relation to the first determination, where the NO concentration in the exhaled air was 126 ppb.

## Data Availability

Data supporting our results are available for request from the corresponding author.

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
