# Peer review of "Pilot Study of Use of Nitric Oxide in Monitoring Multiple Dental Foci in Oral Cavity—A Case Report"

_healthcare, 2022, doi:10.3390/healthcare10020195_

Round 1
Reviewer 1 Report
Dear Authors
The aim of this case report is very interesting, and new. However, the manuscript style is not suitable for accept. Material and methods, and results are not in the text. Please add heading, and correct the configuration.
"2. Case Report" should be under matarials and methods.
In figure 1, "Clinical intraoral apsect."→"Frontal view of patient intraoral in first visit"
L 344: According to the case review→According to this case report
Please add figure of occlusal view before and after extraction. This picture will helpful images for understanding the situation of this patient.
Reviewer 2 Report
Dear authors,
Thank you for submitting the current manuscript.
I hope that my suggestions will help you increase the quality of it.
Lines 24-40 - Abstract is not well organised for a case report as it has the structure of an article.
Line 46 - Required citation
Line 95 - this is not an original figure. requires copyright permission
The structure of the abstract is not similar to the structure of the article which is a case report.
Line 149 - the medical folder of the patient should have been requested in order to establish the presence or absence off the systemic diseases.
Line 165 - please please provide details regarding the type of implants.
Line 344 - Conclusions should be rephrased as they are not supported by the report.
Best regards!
Reviewer 3 Report
The authors report of nitric oxide measurements in exhaled air of a patient in course of dental treatment. Title “Pilot Study of Use of Nitric Oxide in Monitoring Inflammation in a Patient after Implant Treatment—A Case Report” is kind of misleading since patient suffers from multiple dental foci with poor compliance. So, it is not a implant specific study.
All in all, it is to much discussed about periodontitis pathomechanisms and less regarding nitric oxide. Figure 1 is of bad quality.
Did the authors perform multiple measurements or just 1 for each visit? Why didn’t you correlate to a supposed healthy oral status? Please cut number of references or replace and discuss with regard to more airway related pathologies.
English language does need corrections. Reference formatting is not consistent (e.g. page 11, line 486: year is missing; page 11 , line 453: here even month is given and “volume” addressed)
Reviewer 4 Report
The authors have presented an important topic in a well-written scientific manner. However, there are some issues to improve the flow of the manuscript.
Introduction: Please elaborate more on dental implants, on the types and influencing factors for implant-induced inflammation.
Please add the gap in the previous literature and findings and how this manuscript will contribute scientifically and in clinics. Also, a section on the aims and objectives of the studies should be included.
Figure 1 should be clearer and larger in size.
Discussions: Please add the strengths and limitations of this study and future directions.
Round 2
Reviewer 1 Report
The revised manuscript is well managed.
I think that this MS is possible to accept to Healthcare.
Reviewer 2 Report
Dear authors,
Thank you very much for providing the revised version of the manuscript.
Please carefully check the quality of English language as well.
Best regards!
Reviewer 3 Report
I would like to thank the authors for adequate revisions being made. Keep in mimd of continuing study with control group.
